# Repetitive Low-Intensity Vestibular Noise Stimulation Partly Reverses Behavioral and Brain Activity Changes following Bilateral Vestibular Loss in Rats

**DOI:** 10.3390/biom13111580

**Published:** 2023-10-26

**Authors:** Max Wuehr, Eva Eilles, Magdalena Lindner, Maximilian Grosch, Roswitha Beck, Sibylle Ziegler, Andreas Zwergal

**Affiliations:** 1German Center for Vertigo and Balance Disorders (DSGZ), LMU University Hospital, LMU Munich, 81377 Munich, Germany; max.wuehr@med.uni-muenchen.de (M.W.); eva-eilles@web.de (E.E.); magdalena.lindner@med.uni-muenchen.de (M.L.); maximilian.grosch@med.uni-muenchen.de (M.G.); roswitha.beck@tum.de (R.B.); 2Pharmaceutical Radiochemistry, TUM School of Natural Sciences, TU Munich, 85748 Garching, Germany; 3Department of Nuclear Medicine, LMU University Hospital, LMU Munich, 81377 Munich, Germany; sibylle.ziegler@med.uni-muenchen.de; 4Department of Neurology, LMU University Hospital, LMU Munich, 81377 Munich, Germany

**Keywords:** bilateral vestibulopathy, galvanic vestibular stimulation, stochastic resonance, ^18^F-FDG imaging, locomotion, gait ataxia

## Abstract

Low-intensity noisy galvanic vestibular stimulation (nGVS) can improve static and dynamic postural deficits in patients with bilateral vestibular loss (BVL). In this study, we aimed to explore the neurophysiological and neuroanatomical substrates underlying nGVS treatment effects in a rat model of BVL. Regional brain activation patterns and behavioral responses to a repeated 30 min nGVS treatment in comparison to sham stimulation were investigated by serial whole-brain ^18^F-FDG-PET measurements and quantitative locomotor assessments before and at nine consecutive time points up to 60 days after the chemical bilateral labyrinthectomy (BL). The ^18^F-FDG-PET revealed a broad nGVS-induced modulation on regional brain activation patterns encompassing biologically plausible brain networks in the brainstem, cerebellum, multisensory cortex, and basal ganglia during the entire observation period post-BL. nGVS broadly reversed brain activity adaptions occurring in the natural course post-BL. The parallel behavioral locomotor assessment demonstrated a beneficial treatment effect of nGVS on sensory-ataxic gait alterations, particularly in the early stage of post-BL recovery. Stimulation-induced locomotor improvements were finally linked to nGVS brain activity responses in the brainstem, hemispheric motor, and limbic networks. In conclusion, combined ^18^F-FDG-PET and locomotor analysis discloses the potential neurophysiological and neuroanatomical substrates that mediate previously observed therapeutic nGVS effects on postural deficits in patients with BVL.

## 1. Introduction

Bilateral vestibular loss (BVL) causes, amongst other symptoms, postural imbalance during standing and walking, which worsens in darkness and on uneven ground [1,2,3,4]. These deficits may partially improve as patients adapt behavioral strategies that alter multisensory calibration and locomotor control [5,6]. However, postural deficits tend not to dissipate completely over time, which often results in long-term functional impairment [2].

The currently available treatment options for BVL are primarily based on physical therapy that may foster behavioral adaptations to loss of vestibular function [5]. In addition, emerging treatment strategies either aim at restoring peripheral vestibular function (by means of a vestibular implant [7]), substitute residual vestibular input by other sensory sources (e.g., from proprioception [8]), or boost vestibular excitability by an imperceptible vestibular noise stimulation using non-invasive noisy galvanic vestibular stimulation (nGVS) [9,10,11]. The latter principle takes advantage of the fact that a majority of patients with BVL retain some degree of vestibular function [2]. The rationale behind the application of nGVS is stochastic resonance (SR)—a phenomenon according to which (pathologically increased) thresholds for sensory information processing can be lowered by the application of an appropriate amount of low-intensity sensory noise [12,13]. The beneficial therapeutic effects of nGVS, in particular on static and dynamic postural deficits, are well-documented in animal models [14,15,16] and patients with BVL [9,10,11,17,18]. In contrast, it is largely unknown which central nervous networks are altered by nGVS and how these alterations translate into improvements in locomotor and postural performance. Furthermore, it has not yet been examined if nGVS might interact with naturally occurring mechanisms of BVL-induced adaptive neuroplasticity. The latter implies functional and structural changes in multiple brain networks, including the brainstem, cerebellum, thalamus, striatum, sensorimotor cortices, and limbic areas.

To gain a better understanding of the neurophysiological and neuroanatomical substrates underlying the treatment effects of nGVS, the current study investigated regional brain activation patterns and behavioral responses to repeated applications of nGVS in comparison to sham stimulation by serial whole-brain fluorine-18 fluorodeoxyglucose-positron emission tomography (^18^F-FDG-PET) measurements and quantitative locomotor assessments from baseline to 9 weeks after a chemical bilateral labyrinthectomy (BL) in rats. We hypothesized that repeated nGVS would alter neuronal activity in the (sub-)cortical sensorimotor and brainstem–cerebellar networks, which are also engaged in BVL-induced adaptive neuroplasticity. In terms of behavioral effects, we expected transient effects of nGVS on postural performance following BL.

## 2. Methods

### 2.1. Animals and Housing

All rodent experiments were approved by the government of Upper Bavaria and performed in accordance with the guidelines for the use of living animals in scientific studies and the German Law for Protection of Animals (ROB-55.2-2532.Vet_02-21-32). Male Sprague-Dawley rats (N = 17, weight 400 ± 20 g, age 9–10 weeks at the time of surgery, Charles River, Sulzfeld, Germany) were housed two animals per cage in a temperature- and humidity-controlled room with a 12 h light/dark cycle and free access to food and water. All rodents were placed in double-decker cages (GR1800, Tecniplast, Hohenpeißenberg, Germany).

### 2.2. Experimental Procedures

In all rodents, a chemical BL was performed by transtympanic injection of bupivacaine and p-arsanilic acid. This preganglionic vestibular lesion model was selected, as the physiological effect of nGVS relies on the structural integrity of primary vestibular afferent neurons. On days 1, 3, 6, 7, 15, 21, 30, 45, and 60 post-BL, low-intensity vestibular noise stimulation (nGVS at 0.2 mA; N = 8 animals) or sham stimulation (nGVS at 0 mA; N = 9 animals) was administered. Regional brain activation patterns and locomotion performance were assessed before the BL and repetitively until day 60 post-BL. The immediate nGVS treatment (after-)effects on regional brain activations were assessed by the ^18^F-FDG-PET analysis that was conducted subsequent to stimulation on days 1, 3, 7, 15, 30, and 60 post-BL. During tracer uptake, the animals were allowed to move freely for a period of 25 min. PET imaging started 30 min post injection. On each of these days, a concomitant instrumented analysis of locomotion was performed before nGVS treatment to assess the potential long-term effects of the previous stimulations on locomotion. The immediate effects of nGVS on locomotion were assessed on days 6, 21, and 45 post-BL by locomotion analysis directly before and after the nGVS/sham interventions (Figure 1A–C).

### 2.3. Chemical Bilateral Labyrinthectomy

Chemical BL was performed as described previously [19,20]: The animals were anesthetized with 1.5% isoflurane in O_2_ and delivered up to 1.2 L/min via a mask. For surgical analgesia, 1.5 mg/kg of meloxicam was injected s.c. before and 3 days after surgery. An additional 5 mL of saline was injected s.c. as a bolus. After local anesthesia with 1% bupivacaine hydrochloride, a left paramedian incision was made to expose the lamboidal ridge and the external ear canal. The external ear canal was opened just anterior to the exit point of the facial nerve. Then, the tympanic membrane was perforated caudally to the hammer shaft with a 26-gauge needle. About 0.150 mL of a 20% bupivacaine solution was instilled into the tympanic cavity. For about 2 min, the bupivacaine solution was aspirated and instilled slowly again multiple times. The same procedure was followed to instill 0.150 mL of a 10% solution of p-arsanilic acid, which irreversibly desensitized the primary sensory cells of the inner ear [21]. The wound closure was followed by a skin suture, and for preventive antibiosis, 2 mg/kg of marbofloxacin was injected s.c. for three days. This procedure was carried out on both sides, starting on the left side.

### 2.4. Galvanic Vestibular Stimulation

Before stimulation, the rodents were anesthetized with 2% isoflurane in O_2_ (1–2 L/min) via a mask and were kept warm with a heating pad. Two cannulas (26-gauge) were placed bilaterally s.c. in contact with the mastoid process and connected to the electrodes of a constant current stimulator (neuroConn, Illmenau, Germany). The electrical signal consisted of band-limited zero-mean white noise between 0 to 30 Hz [11]. Initially, the signal intensity was gradually increased until an amplitude, at which small involuntary head movements of the animal occurred. The final nGVS intensity was set to 20% of this amplitude, which yielded 0.2 mA across the animals. For each animal and time point of intervention, stimulation was applied for a duration of 30 min.

### 2.5. PET Imaging and Analysis

The rodents were anesthetized with 2% isoflurane in O_2_ (1–2 L/min) via a mask, and a cannula was placed in a lateral tail vein for the ^18^F-FDG bolus injection (in 0.5 mL saline, 40 MBq). Subsequently, the animals were awakened and allowed to move freely for an uptake period of 25 min until anesthesia was induced again with isoflurane (1.8%) for the PET scan (starting 30 min post injection). Two animals per scan were positioned in an Inveon PET scanner (Siemens Healthineers, Erlangen, Germany) and were kept warm with a heating pad. The head was fixed using a custom-made head holder to prevent any passive head movements. Emission data were recorded for 30 min, followed by a 7 min transmission scan using a rotating ^57^Co point source. Upon recovery from anesthesia, the animals were returned to their home cages.

Image processing was performed as described previously (Figure 1D–F) [20,22,23]. The obtained emission measurements were reconstructed using an ordered subsets expectation maximization (OSEM-3D) algorithm with decay, scatter, attenuation, and dead time correction, as well as sensitivity normalization. The resulting images had 212 × 212 × 235 voxels of 0.4 × 0.4 × 0.4 mm^3^. Activity distributions of the ^18^F-FDG scans were used as a surrogate for regional cerebral glucose metabolism (rGCM). The volume containing the brains in the images was cropped and rigidly registered into the PX Rat atlas space (W. Schiffer [24]) using PMOD medical image analysis software (RRID: SCR_016547, v4.004, PMOD Technologies LLC, Fällanden, Switzerland). To achieve comparability, normalization of the whole-brain mean activity was performed after applying a 0.4 mm isotropic Gaussian filter using a brain mask in the atlas space. Subsequently, images were segmented into brain regions using the Px Rat (W. Schiffer) atlas, and in addition, regions of interest for the left and right vestibular nucleus were defined. The rGCM levels of symmetric brain regions were pooled after the preclusion of any statistical side asymmetries. Finally, the mean brain normalized rGCM levels for each of the 26 brain regions were obtained and further statistically processed. Additionally, voxel-wise analysis based on *t*-tests was performed in the SPM 8 software (Wellcome Department of Cognitive Neurology, London, Great Britain, UK) between measurements in the nGVS and sham stimulation groups on days 1, 7, 15, 30, and 60 post-BL for the sake of visualization. For PET data, *p*-values of  <0.001 were considered significant.

### 2.6. Locomotion Analysis

A quantitative assessment of locomotion was performed with the CatWalk system (CatWalk XT, Noldus Information Technology, Wageningen, The Netherlands), which consists of an enclosed walkway (glass plate, 64 cm length) that is illuminated by fluorescent light. Locomotion patterns were recorded optically by a high-speed color camera. Rodents were placed individually into the walkway and allowed to move freely in both directions. Only continuous runs across the walkway with a duration of <5 s and a speed variation of <60% were considered for further analysis [25]. In total, three compliant runs were acquired per animal and assessment time point.

Six spatiotemporal gait parameters linked to balance and coordination were analyzed: locomotion speed (1), and separately for the fore and hind limbs base of support (2), mean and variability of stride length (3 and 4), and of stride time (5 and 6). Variability measures were calculated by means of the coefficient of variation [26].

### 2.7. Statistical Analysis

To assess the impact of BL and the differential effects of post-BL neuroplasticity vs. nGVS on brain activity and locomotion, different univariate and multivariate analyses were performed. Long-term influences of repeated nGVS interventions on the course of post-BL recovery were analyzed by a repeated measures ANOVA with the assessment day and stimulation (nGVS vs. sham) separately on each parameter derived from the locomotion analysis. The immediate effects of nGVS on locomotor performance on the three intervention days were analyzed by a repeated measures ANOVA with a group (nGVS vs. sham), time point (pre- vs. post-nGVS), and intervention day (6, 21, and 45 post-BL). Analogous analyses were performed on the PET scan outcomes (i.e., the normalized rGCM levels) using repeated measures MANOVA that included all analyzed 26 brain regions. Post hoc Bonferroni adjustments were used to control for multiple comparisons within each model. Power estimation was completed by calculating partial eta-squared (η_p_^2^) for each comparison. Based on common conventions, effects were considered medium for η_p_^2^ > 0.06 and large for η_p_^2^ > 0.14. Spearman’s rank correlation analysis was performed to assess the potential associations between intervention-induced changes in brain activity and locomotor performance. The results were considered significant at *p* < 0.05. Statistical analysis was performed using SPSS (Version 26.0, IBM Corp., Armonk, New York, NY, USA).

## 3. Results

### 3.1. Regional Brain Activation Patterns

Chemical BL had an instantaneous impact on regional brain activation patterns (Figure 2, Appendix A). Compared to the baseline assessment pre-BL, the rGCM levels on day 1 post-BL revealed increased activity in 42% of all analyzed brain regions, particularly within the hemispheric sensory and motor networks (e.g., insular, somatosensory, motor cortex, and striatum). A concomitant decrease in activity was found in 19% of the regions, mainly within the brainstem–cerebellar networks (e.g., colliculus inferior and vestibular nuclei).

During the time period from day 1 to 60 post-BL, a steady decrease in rGCM was found in 31% of the brain regions encompassing the hemispheric sensory networks (e.g., auditory and somatosensory cortex), motor networks (e.g., medial prefrontal and motor cortex, as well as striatum), and limbic networks (cingulate and retrosplenial cortex). In contrast, rGCM levels in the brainstem–cerebellar networks (e.g., cerebellar gray matter and inferior colliculus) and the lateral thalamus exhibited an increase during the course of recovery post-BL.

Repeated treatment with nGVS on days 1, 3, 6, 7, 15, 21, 30, 45, and 60 post-BL resulted in differential rGCM responses that, on the whole, reversed the rGCM modulations occurring in the natural course post-BL. Accordingly, the rodents receiving repeated nGVS treatment (compared to sham stimulation) exhibited increased brain activity (Figure 3, Appendix A), in particular in the hemispheric motor networks (e.g., motor cortex and striatum), limbic networks (cingulate and retrosplenial cortex), and a concomitant decrease in the brainstem–cerebellar networks (e.g., cerebellar gray and white matter, colliculus inferior, and vestibular nuclei). These stimulation-induced brain activity modulations were present throughout the entire analyzed 60-day period post-BL.

### 3.2. Locomotor Performance

Analogous to the effects on regional brain activity, chemical BL also directly affected the locomotor capacity of rodents (Figure 4). Compared to the baseline performance, the catwalk assessment on day 1 post-BL revealed a marked slowdown of locomotion linked to a decrease in stride length and a concurrent increase in stride time that was found equally in forelimb and hindlimb locomotion. Acute locomotor changes further exhibited typical features of a sensory-ataxic gait pattern with a broadened base of support and increased variability of the spatiotemporal stepping pattern (i.e., stride length CV and stride time CV) that were more pronounced for hindlimb locomotion.

Impaired locomotion steadily improved within the analyzed 60-day period post-BL. The catwalk assessment on day 60 post-BL revealed a close-to-normal locomotion speed, base of support, and spatiotemporal gait variability levels. Repeated nGVS interventions on days 1, 3, 6, 7, 15, 21, 30, 45, and 60 post-BL only marginally affected locomotor recovery in terms of a decreased forelimb stride length CV on day 30 post-BL (Figure 4D). As a result, the overall course and extent of locomotor recovery following BL were largely comparable between the animals receiving nGVS treatment versus the animals receiving sham stimulation.

Immediate effects of nGVS stimulation on locomotor performance were assessed on days 6, 21, and 45 post-BL (Figure 5). Compared to the sham stimulation, nGVS improved the sensory-ataxic gait alterations of the treated animals in terms of a reduced base of support (Figure 5B) and lowered levels of stride length and time variability (Figure 5D,F). These beneficial treatment effects were only observed on day 6 post-BL when the animals still exhibited marked gait ataxia but not on days 21 and 45 post-BL, when animal locomotion had already largely recovered.

Correlation analysis between the stimulation-induced changes in locomotor performance (on day 6 post-BL) and stimulation-induced regional brain activity (on day 7 post-BL) revealed associations with the brainstem, hemispheric motor, and limbic networks (not corrected for multiple comparisons). The decreased spatial stepping variability correlated with increased rGCM within the vestibular nucleus (R = 0.857; *p* = 0.007) and the decreased temporal stepping variability with elevated rGCM levels in the cingulate (R = 0.821; *p* = 0.012) and retrosplenial cortex (R = 0.714; *p* = 0.036). Moreover, a narrower base of support was associated with increased activity within the motor cortex (R = 0.821; *p* = 0.012).

## 4. Discussion

In recent years, low-intensity vestibular noise stimulation (i.e., nGVS) has shown promising therapeutic effects on sensorimotor deficits in patients with BVL ([9,10,11,17,18,27]) and other disorders associated with central vestibular deficits (e.g., Parkinson’s disease [28,29,30,31]). In this study, we applied whole-brain imaging of glucose metabolism and instrumented locomotor analysis to study the potential neurophysiological and neuroanatomical substrates underlying the aforementioned treatment effects of nGVS in a rodent model of BVL. ^18^F-FDG-PET revealed a broad impact of nGVS stimulation on the regional brain activation patterns encompassing biologically plausible brain networks in the brainstem, cerebellum, multisensory cortex, and motor-basal ganglia circuits. Immediate stimulation-induced modulations of brain activity were consistently present for the entire studied period of 60 days post-BL and broadly reversed brain activity adaptions occurring in the natural course post-BL. A parallel behavioral locomotor assessment revealed a positive impact of nGVS on sensory-ataxic gait alterations in animals’ locomotor patterns that were particularly found during the early stage post-BL. Finally, stimulation-induced locomotor improvements were linked to nGVS brain activity responses in the brainstem, hemispheric motor, and limbic networks. In the following, we will discuss these observations and contrast the general impact of BL and post-BL recovery with the stimulation-induced effects on regional brain activation patterns and locomotor performance.

### 4.1. Impact of Bilateral Labyrinthectomy and Post-Surgery Recovery on Brain Activation Patterns and Locomotor Performance

As reported previously [32], chemical BL resulted in an acute decrease in neuronal activity within the brainstem–cerebellar networks encompassing primary and secondary processing hubs (e.g., vestibular nuclei, inferior colliculus) of the sensory cues from peripheral vestibular and auditory afferents. Decreased regional neuronal activity in brainstem–cerebellar networks only partially recovered during the studied period of 60 days post-BL.

In contrast, we observed a predominant acute increase in regional brain activation in hemispheric sensory, motor, and limbic networks in response to BL. Specifically, neuronal activity in the thalamus—a prominent hub for multisensory integration and vestibular-motor action [33]—further increased during the course of post-BL recovery, which likely reflects processes of adaptive neuroplasticity that facilitate multisensory substitution and recalibration [32].

Alterations of brain activation patterns within frontal-basal ganglia loops following BL substantiate the existing evidence for a considerable connection between the vestibular system and the basal ganglia [34]. Frontal-basal ganglia networks are essentially involved in the supraspinal control of locomotion. In particular, they are considered to facilitate or suppress movement via a direct (excitatory) and indirect (inhibitory) pathway [35]. Altered brain activity within the frontal-basal ganglia networks was further associated with persistent locomotor hyperactivity in the open field in our animals following BL, which corresponds to a well-documented behavioral feature in animal models of BVL [34,36]. This association fits well with the previously proposed hypothesis that a loss of vestibular input to the striatum may enforce basal ganglia output via the direct pathway and thereby activate locomotion [34]. The apparent difference to a previous study that reported decreased instead of increased rGCM levels in frontal-basal ganglia networks following BL might result from methodological differences: While ^18^F-FDG tracer uptake in the previous study was performed in resting rodents, animals in the current study were allowed to move freely during this period. Within the studied period of post-BL recovery, elevated functional activity in the frontal-basal ganglia partially decreased but did, in general, not reach the baseline levels of neuronal activity as assessed pre-BL.

Acute concomitant alterations in locomotor performance were observed following chemical BL. A general slowdown of locomotion was accompanied by a pathognomonic increase in the spatiotemporal variability of the stepping pattern and a broadened base of support, which are both characteristic of a sensory-ataxic gait disorder [3]. Increased spatiotemporal stepping variability likely directly reflects the acute loss of vestibular feedback, which is thought to fine-tune and adapt the spatiotemporal stepping pattern to unintended irregularities occurring during locomotion [37,38]. In contrast, the broadening of the stepping pattern likely represents a secondary consequence of impaired regulation of balance when the body is moving forward. Both gait impairments were considerably more pronounced in the hindlimb stepping pattern, indicating that vestibulo-spinal regulation of balance during quadruped locomotion predominantly governs hindlimb behavior [39,40]. While locomotor impairments steadily improved during the studied recovery period following surgery, moderate signs of gait ataxia on day 60 post-BL indicate a persistent and not fully compensated locomotor deficit.

### 4.2. Vestibular Noise Stimulation Effects on Brain Activity and Locomotor Performance

Low-intensity vestibular noise stimulation is thought to augment residual vestibular function in BVL via SR. We studied the effects of low-intensity nGVS on brain activation patterns on 6 consecutive days, spanning a 60-day period post-BL. Although our experimental setup did not allow for a parallel vestibular stimulation and ^18^F-FDG-PET acquisition, consistent stimulation-induced brain activation patterns were still observed during the 30 min period of tracer uptake following the cessation of stimulation. Prolonged 30 min treatment with nGVS yielded a broad impact on neuronal activity in the brainstem–cerebellar, limbic, hemispheric motor, and sensory networks that were more or less consistent throughout the studied period of post-BL recovery and largely resembled the GVS-induced whole-brain activation patterns previously studied in rats using fMRI [41]. Interestingly, stimulation-induced alterations of brain activity predominantly reversed those observed during the natural course of post-BL adaption. Most prominently, low-intensity nGVS yielded an increase in rGCM within the motor cortex and frontal-basal ganglia loops. This observation is in line with previous reports demonstrating that peripheral vestibular stimulation induces broad neuro-chemical changes at different sites within the basal ganglia [14,15,42,43]. It may further provide a neurophysiological substrate for the observation that treatment with low-intensity nGVS can ameliorate postural symptoms in basal ganglia disorders like Parkinson’s disease [28,29].

Attenuating effects of nGVS on brain activity were observed within the brainstem–cerebellar networks. In particular, in the cerebellum, which is known to contribute to vestibular compensation [23] and exhibited increased activity levels during post-BL recovery, stimulation resulted in lowered levels of activity. This observation could indicate that partial restoration of vestibular input by nGVS might interfere with and interrupt compensatory cerebellar-mediated processes linked to recovery post-BL. Surprisingly, we further observed an nGVS-induced attenuation of neuronal activity at the primary and secondary hubs of central vestibular and auditory processing (i.e., vestibular nucleus and colliculus inferior) that are known to become considerably modulated by GVS [44]. However, GVS-induced responses within brainstem–cerebellar networks have been shown to rapidly habituate and become inhibited over time, which could explain the relative deactivation following a prolonged 30 min treatment with nGVS [44,45].

In parallel to the ^18^F-FDG-PET analysis of nGVS effects on brain activity, we studied the instantaneous impact of stimulation on sensory gait ataxia over three consecutive days spanning a 45-day period of post-BL recovery. Treatment with nGVS, in particular, improved the sensory-ataxic gait alterations in terms of a narrower base of support and reduced levels of spatiotemporal stepping variability. These stimulation effects on animals’ locomotor patterns broadly resemble the previously reported treatment effects of nGVS on locomotor impairments in patients with BVL [10,11]. In accordance with these reports, nGVS-induced improvements in locomotor function were only present for a short interval following cessation of stimulation and did not persist over time. Furthermore, nGVS only impacted locomotion at the early stages of post-BL recovery but not at the later stages, where sensory-ataxic gait deficits had, to a large extent, receded.

nGVS has been shown to facilitate vestibulo-spinal balance responses in healthy humans and patients with BVL [46,47], which may result in the observed stabilization of locomotor performance in ataxic animals post-BL. In line with this assumption, nGVS-induced responses in the vestibular nuclei, which relay motor commands through the vestibulo-spinal tract, were found to be correlated with decreased spatial stepping variability under treatment with nGVS. Beyond improvements at the vestibulo-spinal reflex level, nGVS might also contribute to an attenuation of sensory-ataxic gait deficits via modulation of higher brain network activity. Such a link is indicated by correlations found between decreased temporal stepping variability and a narrower base of support with nGVS-induced responses in the cingulate, retrosplenial, and motor cortex—areas that have been broadly associated with general motor and locomotor control [30,48,49].

### 4.3. Conclusions

The findings of this study should be interpreted in light of certain preconditions. First, our experimental setup did not allow for a parallel vestibular stimulation and PET acquisition or locomotor assessment that were performed immediately after prolonged treatment with nGVS. Hence, we were only able to study the immediate after-effects of prolonged nGVS treatment on brain activity and gait ataxia. Our current data suggest that nGVS induces consistent short-term (i.e., about 30 min) but not long-term (across the intervention days) stimulation after-effects on brain activity and locomotor behavior. Evidence for the presence of such stimulation-induced after-effects on vestibular function is still controversial, with studies reporting prolonged (i.e., lasting for several hours) after-effects on postural stability [27,50] and studies that, in contrast, demonstrate a rapid disappearance of any stimulation effects after cessation of nGVS [51,52]. Nevertheless, the observed stimulation after-effects on brain activation and locomotion closely resemble those observed previously in rodents or humans during active GVS [10,11,14]. Furthermore, in order to robustly test for the presence of nGVS-induced stochastic resonance, previous studies suggested that nGVS effects should be examined across a broad range of stimulation intensities [29,53,54], which was not achievable with our current experimental setup. However, the fixed nGVS intensity applied here was close to the average reported optimal intensity from trans-mastoidal nGVS in humans [9,29,53,55] and the efficacy of stimulation at this intensity is suggested by the close resemblance to the stimulation effects previously reported in humans [10,11,17].

In conclusion, using a rodent model of BVL, we observed that low-intensity vestibular noise stimulation modulates the brain activity encompassing networks of lower and higher vestibular information processing. This stimulation-induced modulation broadly reverses brain activity adaptions that occur during the natural course of post-BL recovery. Furthermore, stimulation effects on hubs of lower and higher central vestibular information processing are linked to improvements in sensory-ataxic locomotor deficits that closely mimic those previously reported in patients with BVL. Hence, the current findings shed light on the potential neurophysiological and neuroanatomical substrates that mediate the previously reported treatment effects of low-intensity vestibular noise stimulation on vestibular perceptual and sensorimotor function in patients with chronic vestibular loss. Low-intensity vestibular noise stimulation in BVL may be a complementary add-on treatment to the current gold standard of vestibular exercises and rehabilitation, as it presumably improves vestibular function for both active and passive movements.

## Figures and Tables

**Figure 1 biomolecules-13-01580-f001:**
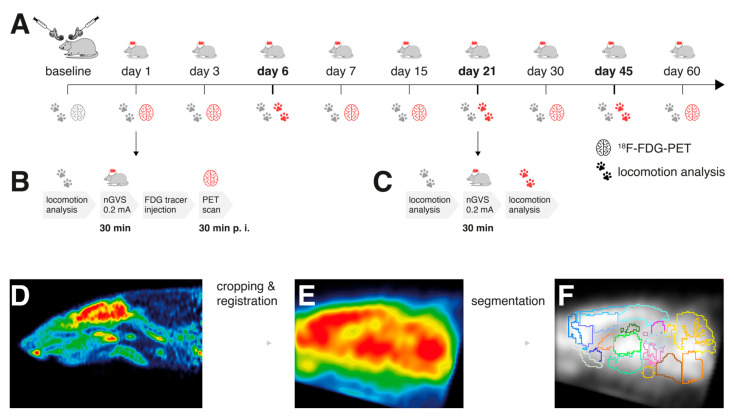
Experimental protocol and processing of brain scans: (**A**) ^18^F-FDG-PET and locomotion analysis were performed before (baseline) and post-bilateral labyrinthectomy (BL) on 9 sequential days encompassing a total period of 60 days. A 30 min long treatment with noisy galvanic vestibular stimulation at 0.2 mA (nGVS; N = 8) vs. sham (N = 9) was applied on each assessment day post-BL. (**B**) On days 1, 3, 7, 15, 30, and 60 post-BL, immediate stimulation effects on regional brain activation were tested by whole-brain ^18^F-FDG-PET imaging directly following nGVS treatment. A concomitant locomotion analysis before nGVS treatment was performed on each of these days to assess potential long-term effects of stimulation. (**C**) Immediate effects of nGVS on locomotion performance were assessed on days 6, 21, and 45 post-BL. (**D**–**F**) Post-processing of PET serial scans: Images were acquired, reconstructed, cropped, registered, filtered, normalized, and segmented into 26 brain regions. Subsequently, activity levels for each brain region normalized to the whole brain were obtained.

**Figure 2 biomolecules-13-01580-f002:**
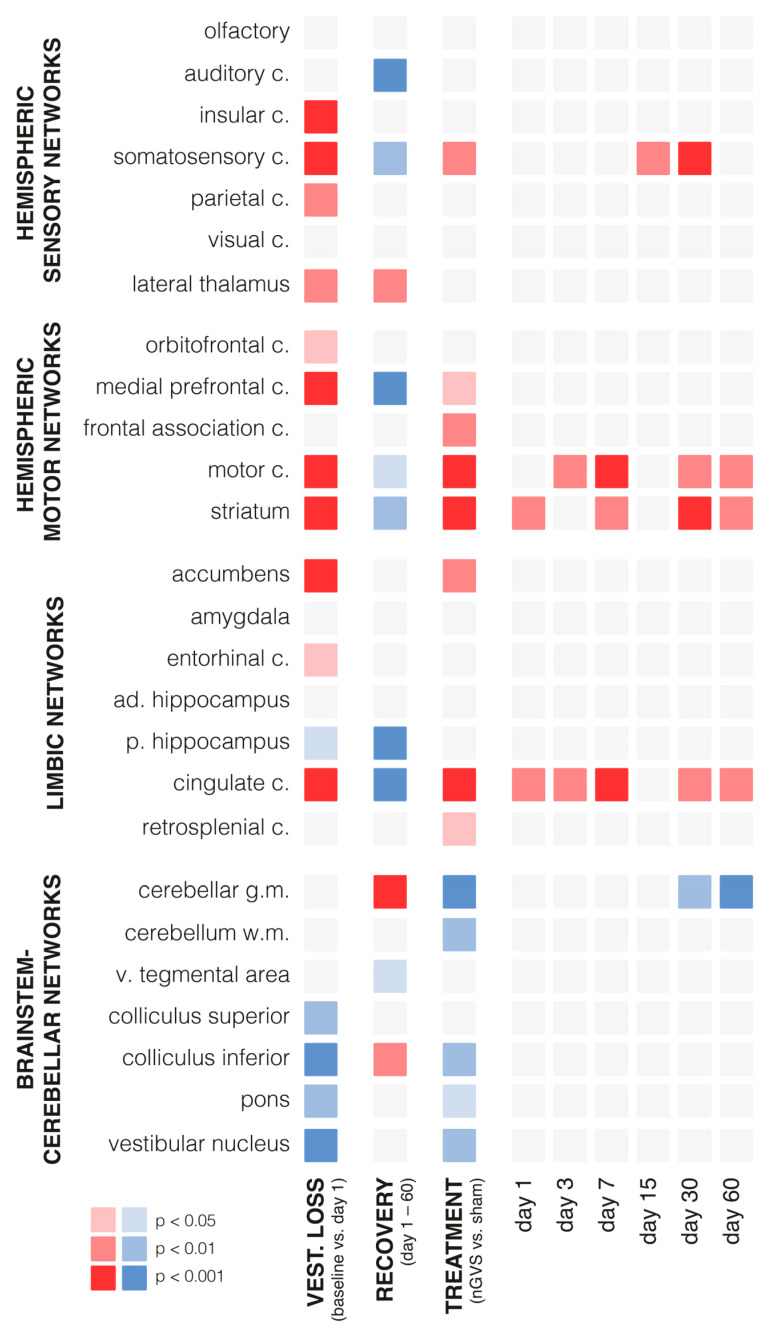
Changes in regional brain activity in response to bilateral labyrinthectomy and low-intensity vestibular noise stimulation. Overview of changes in mean normalized levels of brain activity in 26 selected brain regions measured by ^18^F-FDG-PET. Significant increases in activity are visualized in red (light red: *p* < 0.05; medium red: *p* < 0.01; dark red: *p* < 0.001) and decreases in activity in blue (light blue: *p* < 0.05; medium blue: *p* < 0.01; dark blue: *p* < 0.001). The first column: instantaneous changes in response to bilateral labyrinthectomy (baseline vs. day 1); the second column: changes within the course of recovery (days 1–60 post-bilateral labyrinthectomy); the third column: changes in response to low-intensity vestibular noise stimulation (nGVS vs. sham); the fourth to ninth columns: time-dependent changes in response to vestibular noise stimulation, i.e., decomposition of significant interaction effects between recovery time and treatment. Abbreviations: nGVS: noisy galvanic vestibular stimulation; vest: vestibular; c: cortex; ad: anterior-dorsal; p: posterior; v: ventral.

**Figure 3 biomolecules-13-01580-f003:**
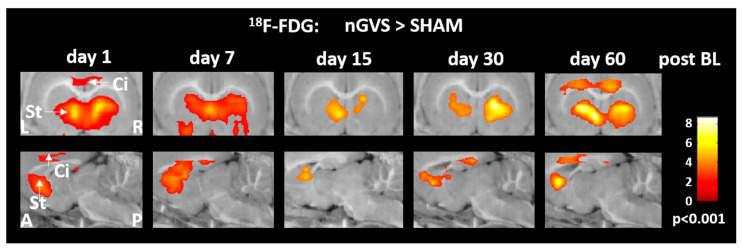
Brain activity responses to low-intensity vestibular noise stimulation in hemispheric motor and limbic networks. Comparison of regional cerebral glucose metabolism in the cingulate cortex and striatum between animals receiving noisy galvanic vestibular stimulation (nGVS) vs. sham stimulation on days 1, 7, 15, 30, and 60 post-BL. The nGVS treatment results in increased activity in both brain regions on all intervention days. Abbreviations: BL: bilateral labyrinthectomy; Ci: cingulate cortex; St: striatum; L: left; R: right; A: anterior; P: posterior.

**Figure 4 biomolecules-13-01580-f004:**
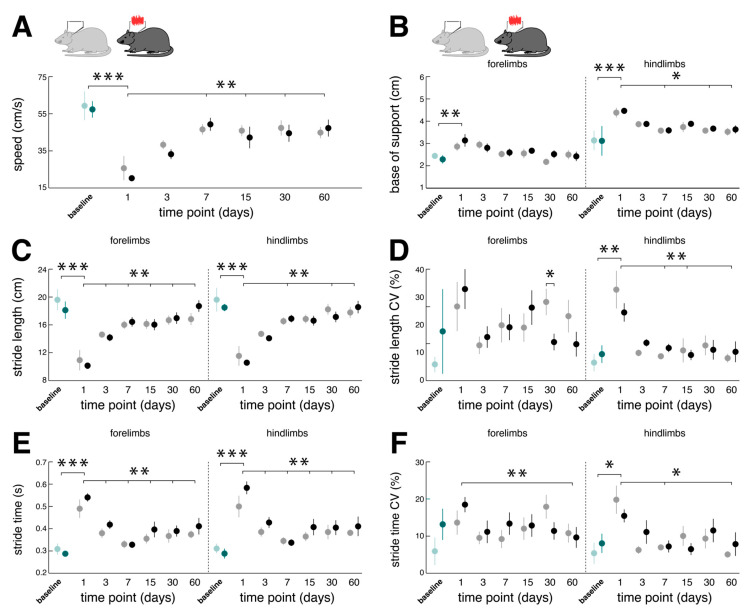
Long-term influences of low-intensity vestibular noise stimulation on locomotor recovery following bilateral labyrinthectomy. Comparison of changes in locomotor performance within a period of 60 days following BL in the nGVS group (dark green and black circles) vs. the sham stimulation group (light green and gray circles). Spatiotemporal gait parameters (**A**–**F**) of locomotor performance assessed on days 1, 3, 7, 15, 30, and 60 post-BL. Locomotion analysis on these days was performed ahead of a 30 min long treatment with nGVS or sham stimulation. Compared to baseline, BL results in a slowdown of locomotion (**A**,**C**,**E**) with a broadened base of support and increased gait variability (i.e., CV; (**D**,**F**)). Locomotor alterations are more pronounced for hindlimbs. Following BL, impaired locomotion performance steadily recovers close to baseline levels. Repeated nGVS interventions only marginally influence the long-term course of locomotor recovery post-BL ((**D**), left column). Abbreviations: nGVS: noisy galvanic vestibular stimulation; BL: bilateral labyrinthectomy; CV: coefficient of variation; * *p* < 0.05; ** *p* < 0.01; *** *p* < 0.001.

**Figure 5 biomolecules-13-01580-f005:**
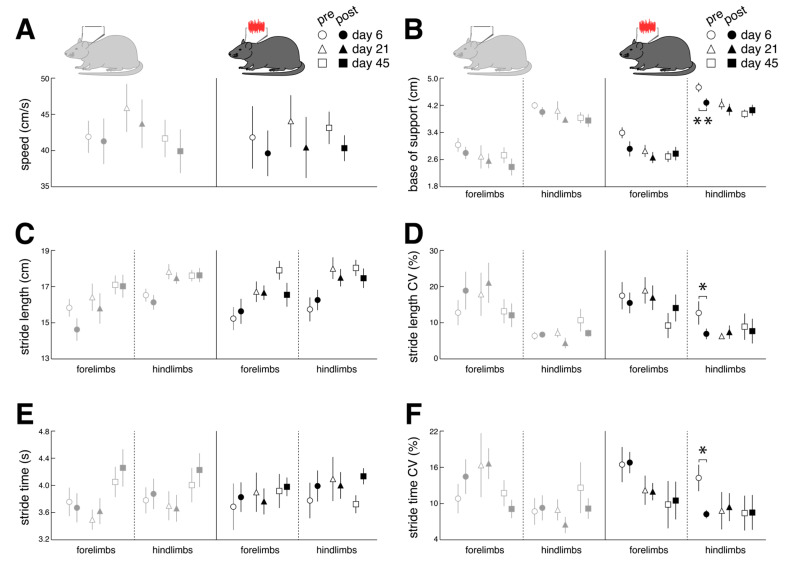
Immediate effects of low-intensity vestibular noise on locomotor performance. Comparison of pre- (open symbols) and post-stimulation (filled symbols) performance with nGVS (N = 8; right column) vs. sham stimulation (N = 9; left column) on intervention days 6 (circles), 21 (triangles), and 45 (squares) on spatiotemporal gait parameters (**A**–**F**). nGVS alters locomotion in terms of a narrower base of support (**B**) and reduced variability (i.e., CV) of stride length (**D**) and stride time (**F**). nGVS effects on locomotion are restricted to hindlimb gait performance and only present during intervention day 6. Abbreviations: nGVS: noisy galvanic vestibular stimulation; CV: coefficient of variation; * *p* < 0.05; ** *p* < 0.01.

## Data Availability

The datasets used and/or analysed during the current study will be available from the corresponding author upon reasonable request.

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
