# Peer review of "Repetitive Low-Intensity Vestibular Noise Stimulation Partly Reverses Behavioral and Brain Activity Changes following Bilateral Vestibular Loss in Rats"

_biomolecules, 2023, doi:10.3390/biom13111580_

Round 1
Reviewer 1 Report
Bilateral vestibular loss (BVL) can lead to postural and gait imbalance, which can be restored with certain behavioral strategies adopted by the patients. The author utilized low-intensity noise stimulation to promote behavioral recovery and observed activation in various brain regions using [18F]-FDG-PET. Most notably, this article observed changes in brain regions associated with behavioral recovery using [18F]-FDG-PET in response to low-intensity noise stimulation. I only have some minor comments:
INTRODUCTION and ABSTRACT:
1. Lack of description of the research background.
2. The introduction section lacks the necessary full form of [18F]-FDG-PET.
METHODS and RESULTS
1. In lines 95 and 143 of the methods resection, ‘p.i.’, I don’t know what that means. It’s a typo? i.p. (Intraperitoneal injection)
2. The author achieved chemical damage through meloxicam. Is this type of damage reversible? There are also studies that achieve labyrinthectomy or neurectomy through physical means. What are the differences between these two methods? Have you considered using [18F]-FDG-PET to observe the overall activation or inhibition status of the regional brain after labyrinthectomy or nerve blockage?
DISCUSSION
1. The author found that low intensity noise stimulation can promote the recovery of posture defects. The principle behind this is that it further enhances recovery through the neuroplasticity of the cortical somatosensory motor and brainstem-cerebellar network. However, during the recovery process, the primary activation is observed in the motor cortex, striatum, and cingulate cortex, while the brain regions crucial for controlling balance, such as the brainstem and cerebellum, do not show significant activation or inhibition. How can we interpret this?
2. There is literature research indicating that exercise training can also promote the recovery of posture defects. In comparison to low-intensity noise stimulation, what are the advantages of the latter?
FIGURE
1. Icons for the experimental group and control group are required in Figure 4 and Figure 5
2. Why does the base of support and stride length in day 15 appear a brief momentary rise of hindlimbs in Figure 4?
Reviewer 2 Report
Line 61: Sentence that starts “In contrast, it is largely unknown 1)…” is clumsily constructed; and removing the 1), 2) and 3) would let this sentence run smoother. Alternatively, ask the journal editor to consider “In contrast, it is largely unknown:” followed by inset numbered lines (shown here for clarity)
1) which central nervous networks are altered by nGVS;
2) how these alterations translate into improvements in locomotor and postural performance; and
3) if nGVS might interact with naturally-occurring mechanisms of BVL-induced adaptive neuroplasticity.
Line 88: “p-arsenic acid” should be p-arsanilic acid.
Reviewer 3 Report
The work describes the results of low-noise galvanic stimulation on central nervous system activation in the presence of bilateral vestibular damage. The evidence of the efficacy of this stimulation support what has already been observed and published with an additional interest on the distinction between brain CNS areas differently involved in vestibular compensation. Despite the undoubted interest of the study, there are some aspects for which the finding of the study is currently not reassuring and sufficiently deepened. The main critical points are:
1. The number of animals tested is rather modest in relation to the fact that chemical lesions of the vestibular system cause deficits of different magnitude and consequently different central activations in the control and electrically stimulated groups. In addition, anesthesia can differently alter the effect of the lesion in the CNS. However, there is no evaluation of the power of statistical analysis to confirm whether the number of animals and experiments was sufficient to overwhelm .the variability
2. The findings are obtained immediately after the electrical stimulation so that the CNS modulation depends on the immediate influence of the stimulus. But, to evidence a contribution of the galvanic stimulation to the improvement of the symptoms and neural responses it would be interesting to show whether galvanic stimulation induces plastic events that leads to persistent changes in the CNS. To that the electrical stimulation should be delivered only once and then the effects of the long-term electrical stimulus should be observed later during the following weeks.
3. The title in my opinion does not clarify the innovative aspects of the research
Round 2
Reviewer 3 Report
I'm sorry but the paper can not be accepted since I had asked that the results of every ANOVA should report the power (eta). If it does not reach sufficient power the value of the analysis is lost.Author Response
Please see the attachment.
